# Analysis of Biochar Addition in Improving Tomato Morpho-Physiological Traits and Water Productivity in Greenhouse

**DOI:** 10.3390/plants14213293

**Published:** 2025-10-28

**Authors:** Abdullah Obadi, Abdulaziz Alharbi, Abdulrasoul Alomran, Abdulaziz G. Alghamdi, Thabit Alqardaeai, Arafat Alkhasha, Hamed A. Al-Swadi

**Affiliations:** 1Department of Plant Production, College of Food and Agricultural Sciences, King Saud University, P.O. Box 2460, Riyadh 11451, Saudi Arabia; aobadi@ksu.edu.sa (A.O.); akhurbi@ksu.edu.sa (A.A.); thabit.alqardaeai@ksu.edu.sa (T.A.); 2Department of Soil Science, College of Food and Agricultural Sciences, King Saud University, P.O. Box 2460, Riyadh 11451, Saudi Arabia; agghamdi@ksu.edu.sa (A.G.A.); aalkhasha@ksu.edu.sa (A.A.); hamidswadi@ksu.edu.sa (H.A.A.-S.)

**Keywords:** deficit irrigation, growth, crop yield, soil amendment, water productivity

## Abstract

Enhancing the utilization of water in desert areas, including Saudi Arabia, has become essential for achieving agricultural sustainability. Biochar effectively mitigates ecological stresses through retaining water, altering soil properties, and providing nutrients for plant growth. This study aimed to examine the impacts of biochar addition on morpho-physiological characteristics, yield, and water productivity (WP) in greenhouses under drought stress conditions. The experiment combined three levels of deficit irrigation (DI)—40, 60, and 80% ETc—as well as two rates of biochar (BC)—BC3% and BC5% *w*/*w* (1.28 and 2.13 kg m^−2^, respectively, for planted area); the control was 100% ETc and untreated soil (BC0%). The results indicated that water deficit hurt the plants’ morpho-physiological characteristics and crop yield. For instance, irrigation water shortage decreased yield by 30.88% at 40% ETc compared to the control (100% ETc). However, DI improved WP by 72.80% at 40% ETc compared to the control. The interaction between DI and BC positively affected morphological, physiological, yield, and WP. For instance, the highest rate of biochar (BC5%) increased yield by 11.92% at 80% ETc compared to untreated plants (BC0%). Similarly, tomato plants treated with 5% biochar under the lowest irrigation level of 40% ETc achieved the highest increase in WP (79.33%) compared to the control (100% ETc and BC0%). In general, DI combined with BC could improve morpho-physiological attributes and yield while increasing the WP of tomato plants in arid and semi-arid regions.

## 1. Introduction

The tomato crop (*Solanum lycopersicum* L.) is considered one of the most widely consumed and popular vegetables worldwide [1]. High tomato production is critical to meet the growing food demand in the Kingdom of Saudi Arabia (KSA). Because tomatoes are so widely consumed, they provide a higher total contribution of essential nutrients to the diet than other vegetables [2]. Tomatoes include many antioxidants, such as vitamin C, phenols, and lycopene [3].

Water scarcity has become a concern for agriculture worldwide. Increasing food production through more effective water use is one of modern agriculture’s most significant difficulties [4,5]. Agricultural development increasingly depends on improving water use efficiency and productivity. The United Nations (UN) estimates that the global population will reach 9.7 billion by 2050, a 32% increase compared to 2015 [6]. This population increase will put more pressure on natural resources and reduce arable land, highlighting the need for modern strategies to enhance productivity without harming biodiversity, water, and soil [7]. The challenge today is to enhance food production while simultaneously reducing water consumption.

Agriculture is the largest freshwater consumer, utilizing 70% of the world’s supply. This issue is particularly pronounced in low- and middle-income countries, where 82% of available water resources are dedicated to farming [8]. In contrast, high-income countries keep agricultural water use to around 30%. This disparity underscores the urgent need for sustainable practices that allow us to feed a growing population without further straining our precious water resources [9]. This makes low-income areas more vulnerable to water shortages due to limited technical and financial capacity to implement modern agricultural and irrigation methods that conserve water and increase crop productivity.

Crop water productivity (CWP) and water use efficiency (WUE) are key sustainable agricultural water management metrics. These indicators help optimize water use to increase crop production while conserving water resources [10]. Therefore, CWP can be defined as the amount of marketable yield to the water consumed by the plant (kg/m^3^) [11,12]. Meanwhile, WUE is defined as the system’s efficiency in providing plant water (the ratio of water available to plants in the root zone to total water applied) [10,13]. Deficit irrigation (DI) reduces water use without significantly lowering yield [14]. However, tomatoes are one of the crops sensitive to drought stress, which impacts growth and yield [5,15]. Adopting water-saving technologies, especially with water-intensive crops such as tomatoes, is crucial for agricultural sustainability. Biochar as a soil amendment and deficit irrigation have enhanced crop productivity and improved water use efficiency. Amending soil with biochar is an effective method for increasing long-term productivity and optimizing the use of water and nutrients [16,17,18,19].

Biochar is a fine-grained organic material with a high carbon content. It is produced from biomass through pyrolysis at temperatures ranging from 300 to 600 °C in an oxygen-limited environment [20,21,22]. Recently, biochar has gained attention due to its potential to enhance WUE by improving soil properties, decreasing evaporation and infiltration, and increasing the soil water-holding capacity [23,24]. Also, biochar improves crop production and WUE by reducing the negative impacts of water scarcity [25]. Numerous studies have demonstrated the advantages of biochar in enhancing soil fertility and crop productivity [25,26,27]. Using biochar enhances the growth of plant shoot and root systems by improving the soil’s physicochemical properties [28]. The biological properties of roots play a significant role in the complete performance of plants through the ability of the plant to obtain water and nutrients [29]. Biochar has many economic, agricultural, and environmental benefits. For example, tomato yield increased by 55.23% compared to untreated soil after biochar application at 50 t/ha [17]. de Castro Filho et al. [28] reported that water deficit (50% of available soil water (ASW)) hurt tomato plant growth. On the other hand, positive impacts were observed on the physiological processes, growth parameters, and biomass of plants growing in soil treated with biochar, regardless of soil water level [22,30]. Biochar application improves soil properties, growth, and productivity of tomatoes. In an experiment with four biochar rates (0%, 1%, 3%, and 5%), results showed significant increases in soil porosity, plant height, yield, and water use efficiency (WUE) with biochar addition [31]. The 5% biochar rate provided the highest yield (54.9 t/ha) and WUE (38.5 kg/m^3^) of tomato plants [32]. Usman et al. [33] investigated biochar’s effect on tomato physiology, yield, and nutritional quality under varying irrigation levels. Two biochar rates (0.1% and 0.2%) and four irrigation levels (100%, 70%, 60%, and 50% field capacity) were applied. Drought stress (50% FC) negatively affected growth and quality, but biochar improved root length, plant height, fruit number, and nutritional content. Adding biochar at 0.2% showed more significant gains than 0.1%, saving 30% water while maintaining tomato fruit yield and nutritional value. Drought negatively affects soil and the production of crops. The results of a study conducted by Ebrahimi et al. [34] on tomato plants using deficit irrigation (100%, 75%, and 50% of plant water requirement (PWR)) with vermicompost and biochar (pistachio and date palm) showed that the combination of vermicompost and pistachio biochar and 100% of PWR resulted in the best growth, yield, and nutrient uptake. Maximum WUE was found with mixed amendments at 50% PWR. Stress metabolites were higher in plants without amendments or with vermicompost only under severe drought. Afaf et al. [35] studied the impact of biochar on soil and plant properties under drought stress. Irrigation levels were set at 100%, 75%, and 45% SWC, with and without biochar (20 g/kg). Biochar increased EC, CEC, pH, and nutrient availability (Ca, K, Mg, Na, Mn) under drought conditions. It also improved leaf area, fresh/dry weight, and productivity, enhancing drought tolerance and promoting the agricultural sustainability of tomato plants.

The combined impact of biochar and deficit irrigation to mitigate the adverse effects of water shortage on tomato crops remains unexplored. We hypothesized that plants exposed to low soil moisture might retain high photosynthetic activity and postpone transpiration inhibition by biochar application. This study aimed to evaluate the effect of applying biochar produced from date palm waste under deficit irrigation on the enhancement of morphological and physiological characteristics, yield, and water productivity of tomato plants under the soil and climate conditions of Saudi Arabia.

## 2. Results and Discussion

### 2.1. Effects of Biochar on Morphological Traits of Tomato Plants Grown Under Water Deficit Conditions

Water shortage negatively affected the tomato plants’ growth parameters, such as stem height and diameter, leaf area, leaf green index (SPAD), and fresh and dry weight of shoots. This could be attributed to insufficient irrigation water to remove salt accumulated in the root zone, which may hinder the plant’s ability to uptake water and nutrients. Depending on the degree and duration of stress, deficit irrigation has been shown to significantly decrease most morphological traits [16,36]. However, biochar addition enhanced all vegetative growth traits (Figure 1). The interaction between soil amendment (biochar) levels and deficit irrigation levels positively impacted all growth traits; compared with untreated plants, biochar addition, specifically 5%, increased all traits under all irrigation levels (Figure 1). The increase in all morphological characteristics can be attributed to biochar’s role in improving soil quality, increasing its water-holding capacity, and stimulating microbial activity in the root zone [37]. Additionally, biochar’s high content of minerals, such as nitrogen, magnesium, calcium, and inorganic carbon, significantly contributes to plant growth [38]. Research has indicated the efficiency of biochar in enhancing soil structure [39] by improving porosity, surface area, bulk density, water-holding capacity (WHC), ion exchange capacity, and nutrient use efficiency and availability, thus increasing crop growth and productivity [40,41,42,43]. The success of biochar depends on the biochar type, application rate, and soil conditions [44]. In addition to the benefits of using biochar in soil, deficit irrigation with biochar further enhanced tomato growth [16,26]. Moreover, adding biochar at a rate of 4% (*w*/*w*) increased the fresh and dry weight of shoots, plant height, and stem diameter of tomato plants [45].

### 2.2. Effects of Biochar on Physiological Processes of Tomato Plants Grown Under Water Deficit Conditions

Leaf gas exchange (photosynthetic rate (Pn), the transpiration rate (TR), the conductivity (Cond)), and LRWC were negatively affected by water deficit, particularly when plants were subjected to 40 and 60% ETc compared to 80 and 100% ETc. Regarding the proline content, the highest water stress (40% ETc) resulted in the highest proline content in tomato plant leaves (Figure 2). This could be attributed to the inability of tomato plants to obtain sufficient water from the soil to meet the demands of evapotranspiration, thus increasing the plant canopy temperature, which resulted in decreased stomatal conductance. Compared to well-irrigated plants, stomatal conductivity in tomatoes grown under deficit irrigation conditions has been found to decrease from 14% to 73%, depending on the severity and duration of water stress [46,47]. The Pn decline was distinctly parallel to the decrease in conductivity and TR with the water stress [48,49,50,51]. Liang et al. [51] showed that water stress significantly inhibited the photosynthesis process in tomato leaves. Hao et al. [52] reported that water stress negatively affected photosynthesis parameters and stomatal traits and increased proline content in tomato leaves. Al-Harbi et al. [53] and Alhoshan et al. [54] reported that proline content increased significantly due to deficit irrigation and that the increase in proline content was associated with increased drought. However, adding biochar increased leaf gas exchange properties and LRWC and lowered proline content compared to untreated plants (Figure 2).

Adding organic soil amendments (biochar) reduced the negative effect of water shortage. Under all irrigation water levels, 5% biochar treatment gave the highest significant mean values for leaf gas exchange properties and LRWC, followed by 3% biochar treatment. Conversely, biochar-treated plants showed lower proline content, especially those that were fully irrigated (100% ETc) (Figure 2). The increase in gas exchange in the leaves may be attributed to biochar’s ability to improve soil properties and plant growth parameters. Biochar significantly increased plant growth and all physiological traits under deficit irrigation [55]. According to Abdelghany et al. [56], when tomato plants were subjected to a water deficit, the biochar addition significantly increased the Pn and RWC and decreased the proline content. Gharred et al. [57] obtained similar results and found that biochar significantly increased the leaf gas exchange and transpiration rates of both stressed and non-stressed tomato plants. Moreover, under water shortage, biochar application enhanced the LWRC and gas exchange in leaves, indicating that biochar helped plants maintain robust leaves in the face of abiotic stresses [58].

### 2.3. Effects of Biochar on Photosynthetic Pigments of Tomato Plants Grown Under Water Deficit Conditions

Compared with well-irrigated plants (80 and 100% ETc), photosynthetic pigment-related traits (chlorophyll a, chlorophyll b, total chlorophyll, and carotenoids) showed a decrease, especially at high water deficits of 40 and 60% ETc. On the contrary, biochar application enhanced all photosynthetic pigments in the leaves of treated plants compared to untreated plants (Figure 3). Similarly, the interaction between soil amendments (biochar) and deficit irrigation levels increased all photosynthetic pigment traits of tomato leaves. For instance, the highest percentage of biochar (B 5%) gave the highest values of all photosynthetic pigment traits under all irrigation levels compared to untreated plants (BC0%) (Figure 3). The decreased chlorophyll content could be attributed to damage to thylakoid membranes and membrane permeability, resulting from the destructive effect of reactive oxygen species (ROS) on chloroplasts [57]. ROS production increased in response to a water shortage [59,60]. Our results agree with [61,62,63], which showed that biochar addition increased chlorophyll content under drought stress conditions. Based on our findings in this study, biochar addition enhances the photosynthetic rate, which is an indicator of increased chlorophyll. In addition, Qian et al. [64] observed a significant increase in the chlorophyll levels (41%) of plants grown on biochar-amended soil compared to unamended soil. Also, Younis et al. [65] reported increases in the chlorophyll content (29% for chlorophyll a, 52% for chlorophyll b, and 33% for total chlorophyll) and carotenoid (5%) when plants were grown on soil that received biochar and was subjected to moisture stress.

### 2.4. Effects of Biochar on Total Yield (kg m^−2^) and WP (kg m^−3^) of Tomato Plants Grown Under Water Deficit Conditions

Total yield and irrigation water productivity were significantly affected by deficit irrigation and soil amendments (biochar) (Table 1). Irrigation water shortage decreased total yield by 5.91%, 16.16%, and 30.88% at 80%, 60%, and 40% ETc, respectively, compared to the control (full irrigation at 100% ETc). Several research results have shown that different levels of deficit irrigation reduced the tomato yield from 15.52% to 51.99% compared to full irrigation [26,66,67]. On the other hand, deficit irrigation improved water productivity by 17.61%, 39.76%, and 72.80% at 80%, 60%, and 40% ETc, respectively, compared to the control. Biochar addition increased yield by 3.75% and 6.39% at BC3% and BC5%, respectively, compared to untreated plants (BC0%). Also, irrigation water productivity improved by 3.61% and 5.91% at BC3% and BC5%, respectively, compared to BC0%. Similar results were reported by Akhtar et al. [68], who showed a significant increase in the yield and water productivity of tomato plants with biochar application. Also, Zhang et al. [26] found that biochar increased tomato yield by 30.92% compared to untreated plants. WP increased with biochar additions at rates of 10, 20, 30, and even 40 t/ha^−1^ and decreased at higher application rates of 60 t/ha^−1^, according to a two-year field experiment with tomato plants [69].

The interaction between deficit irrigation and biochar positively affected yield and water productivity (Figure 4). For instance, the highest rate of biochar (BC5%) increased yield by 4.59%, 11.92%, 4.96%, and 3.50% at 100%, 80%, 60%, and 40% ETc, respectively, compared to untreated plants (BC0%). Similarly, tomato plants treated with 5% biochar under the lowest irrigation level of 40% ETc achieved the highest increase in WP (79.33%) compared to the control (100% ETC and 0% BC). The increased yield and WP under water stress may be due to biochar’s capacity to retain water, improve porosity, and provide plant nutrients. WP was also increased by stomatal closure and decreased transpiration rate (TR) in response to water stress [16,70]. Adding biochar enhanced the soil’s physical properties, resulting in higher soil moisture content and better water transport and distribution in the root zone, thus improving WP and yield [71,72]. Diedhiou et al. [73] reported similar results; they found that biochar addition under deficit irrigation increased the yield and WP of tomato plants compared to full irrigation without biochar. Guo et al. [74] found that the WP and yield of tomato plants increased by 45.33% and 55.23%, respectively, with biochar applied at 50 t/ha. According to studies by Obadi et al. [22] and Akhtar et al. [68], biochar amendment improves the WP and yield of tomato plants exposed and not exposed to water stress.

### 2.5. Selection of the Best Treatments (IRBC) Through Their Association with the Studied Traits

The heatmap analysis divided the treatments into three main groups based on the results of the interaction between irrigation levels and biochar (IRBC) treatments on the traits studied. The first group (IR80BC3, IR100BC5, IR100BC0, IR100BC3, and IR80BC5) showed the highest values for most studied traits (red colors), followed by the second group (IR80BC0, IR60BC3, and IR60BC5), which gave medium values (yellow colors). The third group (IR40BC0, IR60BC0, IR40BC3, and IR40BC5) showed the lowest values for all parameters (blue colors) except for proline and WP (Figure 5).

## 3. Materials and Methods

### 3.1. Location and Design of Experiment

From September 2016 to April 2017, a greenhouse experiment was conducted at Al-Mohous Farm in Thadiq Prefecture, 120 km northwest of Riyadh City, Saudi Arabia. The farm’s coordinates are 25°17′40″ N and 45°52′55″ E, and it is located at an altitude of 600 m above sea level (Figure 6). Before the experiment was conducted, samples of irrigation water and greenhouse soil were taken. The chemical analyses of the irrigation water, shown in Table 2, were performed according to Maiti [75]. At the same time, the physical and chemical analyses of the soil samples were performed according to the method of Sparks et al. [76] (Table 2).

The experiment was conducted in a split-plot and randomized complete block design with three replications. The main plots were allocated to three levels of deficit irrigation water—40, 60, and 80% evapotranspiration (ETc)—with full irrigation (100% ETc) acting as the control. The subplots were randomly distributed to different levels of soil amendment—3 and 5% (*w*/*w*) biochar (denoted as BC3% and BC5%)—with no biochar acting as the control. The experiment included 36 units: 4 irrigation levels × 3 biochar levels × 3 replicates. Each experimental unit consisted of a 6 m long and 1 m wide line. The irrigation line spacing was 0.40 m between emitters, with a total of 15 plants. The control treatment was defined as 100% ETc without biochar additives (Figure 7). Measurements were performed on 5 randomly selected plants from each experimental unit.

Under controlled conditions in a polyethylene film-covered greenhouse, the average greenhouse temperature was kept at 25 ± 2 °C and 20 ± 2 °C during the day and night, respectively, and the relative air humidity was 75 ± 2% during the growth periods. On 19 September 2021, commercial tomato (*Solanum lycopersicum* L.) seeds (Tone Guitar, hybrid tomato) were sown in foam pots filled with vermiculite and peat moss (1:1 *v*/*v*). The seedlings were transferred to the greenhouse four weeks after planting. All recommended agricultural practices under commercial greenhouse conditions for tomato production were applied, including pest control, soil sterilization, and fertilization, according to the recommendations of local farmers in the study area (285 kg N, 142 kg P, and 238 kg K per ha), and micronutrients were added by spraying.

The type A evaporator pan installed on bare soil was used to estimate daily evapotranspiration inside the greenhouse according to FAO-56 recommendations [77]. Based on the values of daily evapotranspiration and crop coefficient (Kc), the irrigation levels of 40, 60, 80, and 100% ETc were determined according to Allen et al. [77] through the following equation: *ETc* = *Eo* × *Kp* × *Kc*(1)
where ETc is crop evapotranspiration in mm day^−1^, K_P_ is the pan coefficient, Eo is the evaporation from pan A in mm day^−1^, and Kc is the crop coefficient. The pepper crop underwent four different growth stages: initial stage, 30 days; development stage, 50 days; mid-season stage, 135 days; and late-season stage, 33 days. Different Kc values were used for each stage (0.60, 1.22, and 0.80 for the early stage, mid-season, and late season, respectively), as specified by Allen et al. [77].

### 3.2. Biochar Preparation

Biochar produced from palm waste in Al-Mahous farms, 120 km northwest of Riyadh, was used in this study. Palm fronds were collected, sun-dried, cut into small sizes (10–15 cm), then packed in an oven. The oven was a cylindrical stainless-steel container that was tightly covered to provide oxygen-free conditioning. Pyrolysis was carried out at 450 ± 20 °C. The biochar was ground and sieved before being mixed with greenhouse soil at different rates (3 and 5%) to a depth of 15 cm and a width of 20 cm per m^2^ (1.28 and 2.13 kg m^−2^, respectively, for the planted area), as shown in Figure 7. More details about biochar production from palm waste can be found in [22,78]. Using Micromeritics ASAP 2020 BET, the surface area was calculated. The pH and electrical conductivity (EC) of an aqueous extract prepared from biochar at 1:10 (*w*/*v*) were measured. The amounts of carbon (C), nitrogen (N), and hydrogen (H) were determined using a CHN analyzer (Series II; Perkin Elmer, Waltham, MA, USA). According to ASTM D1762-84, the moisture, fixed carbon, mobile matter, and ash content of biochar were calculated [79]. Table 3 shows the chemical and physical properties of the biochar.

### 3.3. The Measurements

#### 3.3.1. Growth and Physiological Parameters

Vegetative growth traits were measured, including leaf area using an LI-COR device (model 3000A Li-Cor, Bioscience, Lincoln, NE, USA) and plant height and stem diameter using tape measurement. Meanwhile, the fresh and dry weight of shoots (stems and leaves) was determined using a digital weighing balance by weighing the plant cut directly from the soil surface (fresh weight) and then weighing after drying at 75 °C until constant weight using a forced-air oven (dry weight). The leaf relative water content (LRWC) was calculated using Equation (2). The leaves were sampled in the form of discs to obtain the fresh weight. Then, the discs were immersed in deionized water for up to 4 h to obtain the swollen weight. The discs were dried in a forced-air oven at 75 °C until the weight was constant to obtain the dry weight [80].
(2)$$LRWC=\frac{fresh\;weight-dry\;weight}{swollen\;weight-dry\;weight}*100$$

Photosynthetic rate (Pn), conductance (Cond), and transpiration rate (Tr) of tomato leaves were measured using a portable photosynthesis meter (Li-Cor, Lincoln, NE, USA) on three tomato plants per experimental unit. The photosynthetic pigment content of leaves (mg/g fresh weight), including chlorophyll a (Chl. a), chlorophyll b (Chl. b), total chlorophyll (TChl), and carotenoids (CARs), was determined spectrophotometrically (T 80 UV/Visible Spectrophotometer, PG Instruments Ltd., Lutterworth, UK) according to [81]. Chlorophyll a, chlorophyll b, total chlorophyll, and carotene were determined by Equations (3), (4), (5), and (6), respectively, according to [82]. The proline content of leaves was estimated using Clausen’s method [83].
(3)$$Chl.a=\left[\left(12.7*OD\;663\right)-\left(2.69*OD\;645\right)\right]*\frac{V}{1000}*W$$(4)$$Chl.b=\left[\left(22.9*OD\;645\right)-\left(4.68*OD\;663\right)\right]*\frac{V}{1000}*W$$(5)$$TChl=[\left(20.2*OD\;645+\left(8.02*OD\;663\right)\right]*\frac{V}{1000}*W$$(6)$$CARs=\left[OD\;480+\left(0.114*OD\;663\right)\right]*\left(0.638*OD\;645\right)$$
where OD is the extract’s optical density at the wavelength shown, V is the extract’s volume (mL), and W is the fresh weight of leaves (g)

#### 3.3.2. Total Yield and WP

Total fresh fruit yield (TFFY) was calculated by weighing the total fruit harvested from each experimental unit using a digital balance throughout the harvest period and expressed as kg m^−2^ using Equation (7), according to Foley et al. [84]. Crop water productivity (CWP) is the total fresh fruit yield (TFFY, kg m^−2^) divided by the total irrigation water applied (TIWA, m^3^ m^−2^) throughout the growing season on tomato plants, according to [11,12]:(7)$$TFFY=\frac{Yield\;(kg)}{Area\;{(m}^{2})}=kgm^{-2}$$(8)$$CWP=\frac{TFFY\;{(kg\;m}^{-2})}{TIWA\;{{(m}^{3}\;m}^{-2})}=kgm^{-3}$$

According to [85], the yield reduction ratio (YR%) and water saving (%) were calculated using Equations (9) and (10), respectively. Equations (11) and (12) were used to calculate WP and yield improvement, respectively, as described in [18].(9)$$YR\left(\%\right)=\left[\frac{\left(yield\;of\;control-yield\;of\;treatment\right)}{yield\;of\;control}\right]*100$$(10)$$Water\;saving\;(\%)=\left[\frac{\left(WCC-WCT\right)}{WCC}\right]\times 100$$(11)$$Improve\;Yield\;\left(\%\right)=\left[\frac{\left(yield\;of\;treatment-yield\;of\;control\right)}{yield\;of\;control}\right]*100$$(12)$$Improve\;WP\;\left(\%\right)=\left[\frac{\left(WP\;of\;treatment-WP\;of\;control\right)}{WP\;of\;control}\right]*100$$
where WCC and WCT represent the water consumption for the control and treatment (m^3^/m^2^), respectively.

### 3.4. Statistical Analysis

The data analysis was conducted using the SAS software (Version 2020). The study data were statistically compared using analysis of variance (ANOVA), and the least significant difference (LSD) test was applied at a 0.05 confidence level, according to [86].

## 4. Conclusions

Irrigation water shortage is a pressing challenge in Saudi Arabia, primarily due to the country’s arid climate and the scarcity of available water resources. The predominantly sandy soil in this region further complicates water retention, leading to increased evaporation rates and decreased agricultural productivity. As a result, finding sustainable solutions to improve water usage in agriculture is critical for enhancing food security. One promising approach to addressing these agricultural challenges is using biochar as a soil amendment. Biochar is a stable, carbon-rich material produced from biomass pyrolysis, and it has been shown to improve soil properties and plant growth. In our study, we investigated the effects of biochar on the growth and yield of tomato plants cultivated under greenhouse conditions, specifically focusing on how it affects morpho-physiological traits and water productivity (WP) under different levels of deficit irrigation.

Our results demonstrated that the addition of biochar significantly enhanced various growth parameters of tomato plants. For instance, when applying the highest biochar rate of 5%, we observed yield increases across all tested irrigation levels, with improvements of 4.59% under full irrigation (100% of crop evapotranspiration, ETc), 11.92% at 80% ETc, 4.96% at 60% ETc, and 3.50% at the most severe level of 40% ETc compared to untreated plants (0% biochar). These findings indicate that biochar helps mitigate yield losses associated with water scarcity and supports better overall plant health. In addition to the improvements in yield, we recorded a remarkable increase in water productivity. At the lowest level of irrigation (40% ETc), applying biochar at a 5% rate led to a substantial increase in WP by 79.33% compared to the control group. This indicates that biochar not only enhances the ability of plants to grow under limited water availability but also contributes to more efficient water usage.

Given these findings, we strongly recommend the application of biochar to sandy soil as an effective strategy to improve tomato plants’ growth, yield, and water productivity. This approach is especially crucial in water-scarce regions of Saudi Arabia, where optimizing water resources is essential for sustainable agriculture and food security. Farmers can potentially overcome some of the challenges posed by the region’s harsh climatic conditions by integrating biochar into agricultural practices.

## Figures and Tables

**Figure 1 plants-14-03293-f001:**
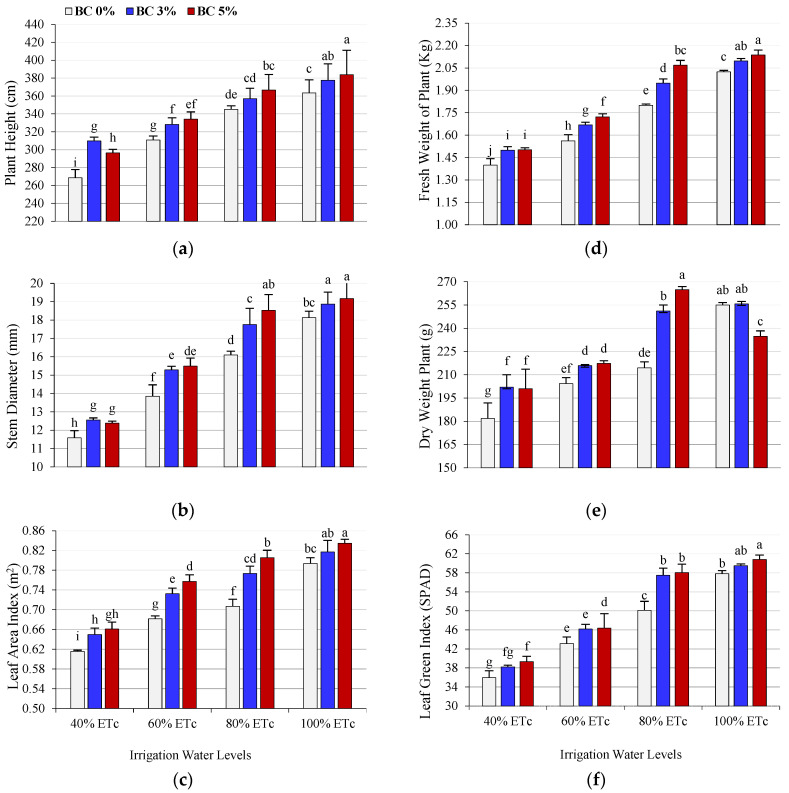
Effects of interaction between deficit irrigation water levels (DI) and biochar (BC) on morphological characteristics of tomato plants, such as plant height (**a**), stem diameter (**b**), leaf area (**c**), fresh shoot weight (**d**), dry shoot weight (**e**), and Leaf Green Index (**f**). According to the LSD test at the 0.05 probability level, means with the same letters do not differ significantly.

**Figure 2 plants-14-03293-f002:**
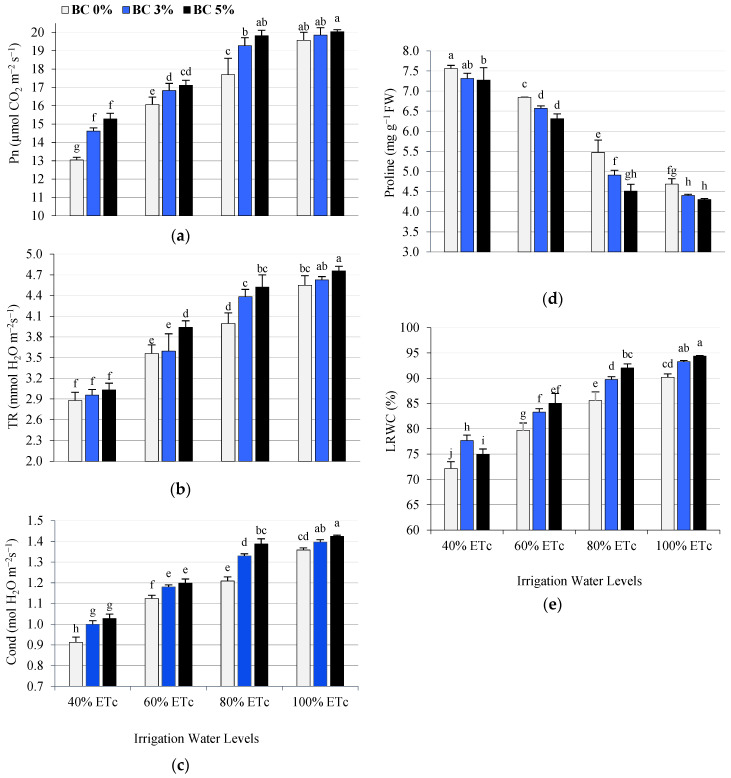
Effects of interaction between deficit irrigation water levels (DI) and biochar (BC) on physiological parameters of tomato plants, such as the photosynthetic rate (Pn) (**a**), the transpiration rate (TR) (**b**), the conductivity (Cond) (**c**), the proline (**d**), and the relative water content (LRWC) of tomato leaves (**e**). According to the LSD test at the 0.05 probability level, means with the same letters do not differ significantly.

**Figure 3 plants-14-03293-f003:**
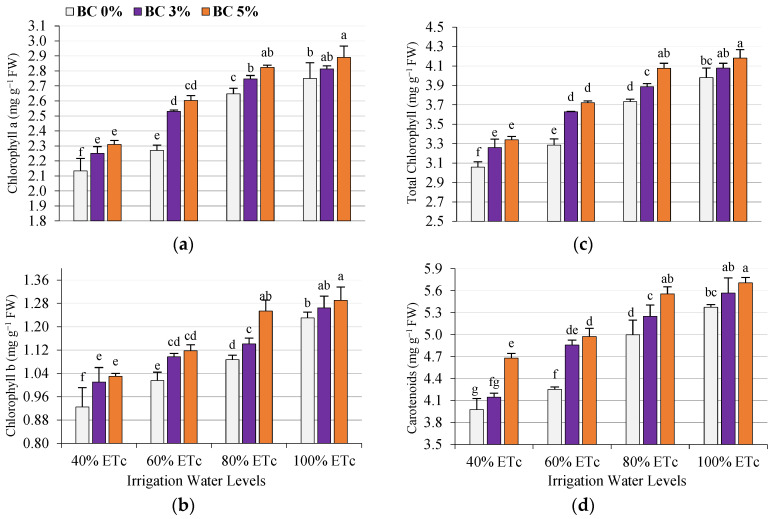
Effects of the interaction between deficit irrigation water levels (DI) and biochar (BC) on the photosynthetic pigments of tomato plant leaves, such as Ca (**a**), chlorophyll b (**b**), total chlorophyll (**c**), and carotenoids (**d**). According to the LSD test, at the 0.05 probability level, columns that have the same letter are not substantially different.

**Figure 4 plants-14-03293-f004:**
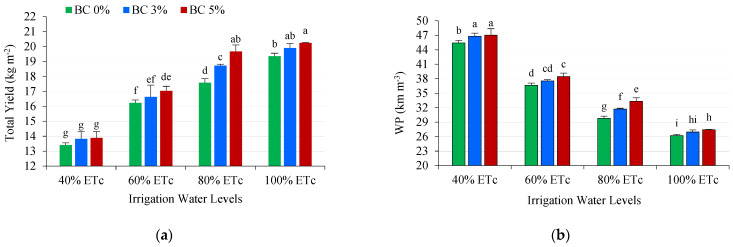
Effect of interaction between deficit irrigation water levels (DI) and biochar (BC) on yield kg/m^2^ (**a**) and water productivity kg/m^3^ (**b**). According to the LSD test, means with the same letters do not differ significantly at the 0.05 probability level.

**Figure 5 plants-14-03293-f005:**
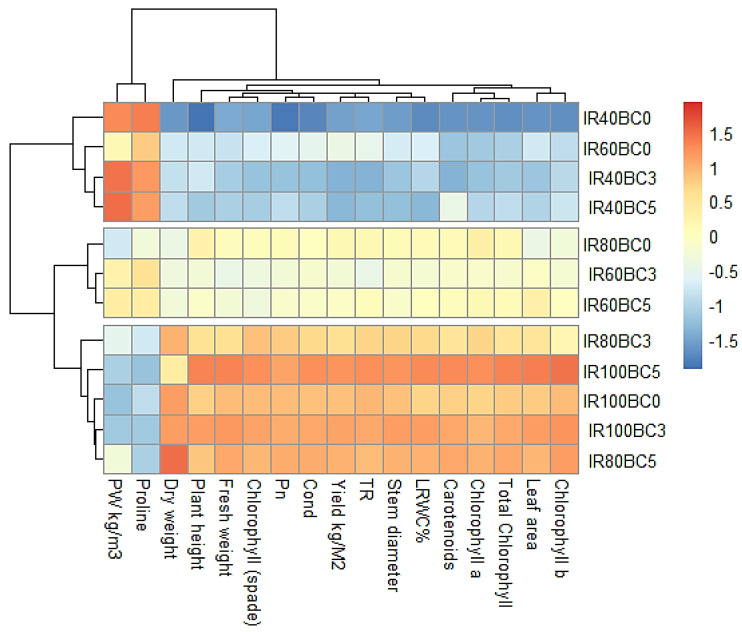
Heatmap dividing the treatments (IRBC) by their association with the traits studied. WP: water productivity, Pn: the photosynthetic rate, Cond: the conductivity, TR: the transpiration rate, and LRWC: the leaf relative water content.

**Figure 6 plants-14-03293-f006:**
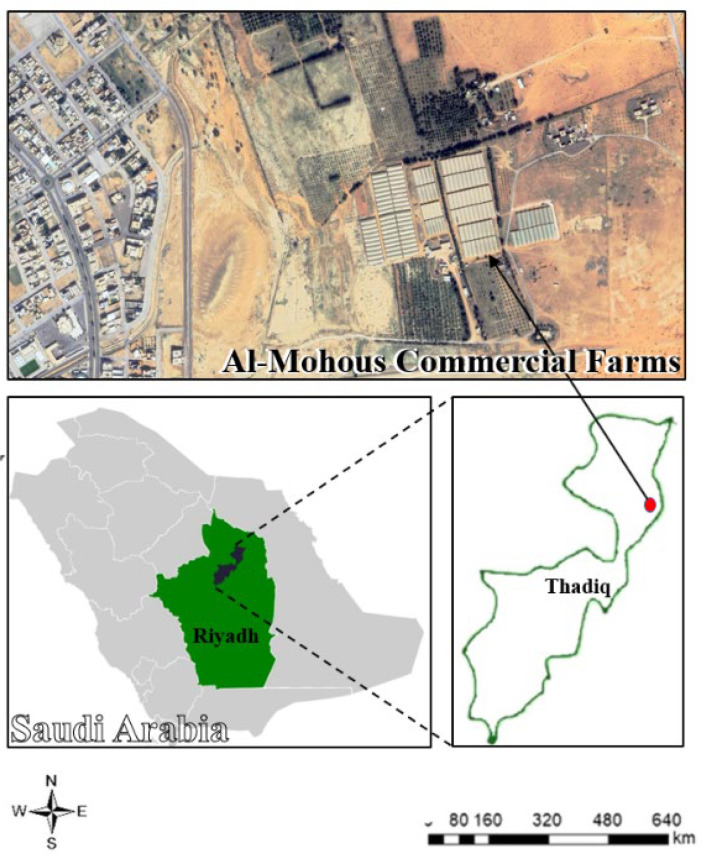
Experiment site: Thadiq Governorate, Riyadh, Kingdom of Saudi Arabia.

**Figure 7 plants-14-03293-f007:**
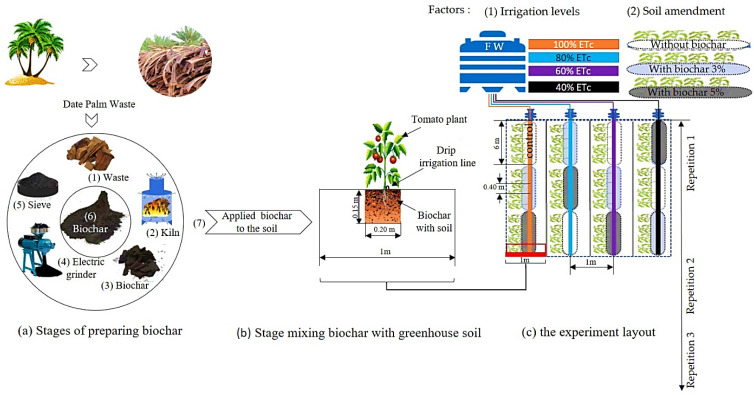
Sketch showing (**a**) stages of preparing biochar, (**b**) mixing biochar with greenhouse soil, and (**c**) the experiment layout and randomization of the treatments.

**Table 1 plants-14-03293-t001:** Impact of deficit irrigation (DI) and biochar (BC) on total yield (kg m^−2^), irrigation water productivity (kg m^−3^), yield reduction and improvement, and water saving of tomato plants.

Treatments	Total Water Applied (m^−3^ m^−2^)	Total Yield (kg m^−2^)	Reduction in Yield (%)	WP (kg m^−3^)	Improvement in WP (%)
**Irrigation Levels (%Etc)**					
**100**	0.74	19.83 ± 0.43 a	0.00	26.87 ± 0.58 d	0.00
**80**	0.59	18.66 ± 0.95 b	5.91	31.60 ± 1.60 c	17.61
**60**	0.44	16.62 ± 0.56 c	16.16	37.55 ± 0.90 b	39.76
**40**	0.30	13.70 ± 0.39 d	30.88	46.42 ± 1.09 a	72.80
**Biochar**			**Improvement in Yield** **(%)**		
**BC0%**	-------	16.64 ± 2.27 c	0.00	34.51 ± 7.67 c	0.00
**BC3%**		17.26 ± 2.45 b	3.75	35.76 ± 7.75 b	3.61
**BC5%**	-------	17.70 ± 2.65 a	6.39	36.55 ± 7.55 a	5.91

According to the LSD test, means with the same letters do not differ significantly at the 0.05 probability level.

**Table 2 plants-14-03293-t002:** Chemical and physical parameters of irrigated water and soil at the experimental location.

**Chemical Analysis**										
**Parameters**	**pH**	**EC (dSm^−1^)**	**Cations (meql^−1^)**				**Anions (meql^−1^)**			**SAR**
			**Ca^2+^**	**Mg^2+^**	**K^+^**	**Na^+^**	**CO_3_^2−^**	**Cl^−^**	**HCO_3_^−^**	
**Water**	7.51	0.94	3.20	2.56	0.14	4.71	0.00	7.92	2.33	2.02
**Soil**	7.38	3.64	10.04	2.24	9.10	13.78	0.00	4.55	18.4	2.03
**Soil Physical Parameters**										
**Parameters**	**ρb** **(g cm^−3^)**	**CaCO_3_** **(%)**	**OM** **(%)**	**Mechanical Analysis (%)**				**θ_s_** **%**	**θ_fc_** **%**	**θ_WP_%**
				**Sand**	**Silt**	**Clay**	**Soil Texture**			
**Soil**	1.42	18.8	0.7	85.5	17.9	7.1	**Loamy sand**	24.8	17.9	9.7

Bulk density (ρb), organic matter (OM), saturated water content (θs), field capacity (θFC), wilting point (θWP), acidity or basicity solution of water and soil (pH), electrical conductivity (EC), and sodium adsorption ratio (SAR).

**Table 3 plants-14-03293-t003:** Physicochemical properties of biochar.

Parameters	Moisture (%)	Resident Material (%)	SA m^2^ g^−1^	OM %	Ash %	pH	EC dS m^−1^	N %	P %	K %	C %	H %	Na %	C/N
**biochar**	3.53	47.90	237.80	30.33	25.70	8.82	3.71	0.24	0.22	0.88	60	3.44	5.63	250:1

## Data Availability

The original contributions presented in this study are included in the article. Further inquiries can be directed to the corresponding author.

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
