# Peer review of "Analysis of Biochar Addition in Improving Tomato Morpho-Physiological Traits and Water Productivity in Greenhouse"

_plants, 2025, doi:10.3390/plants14213293_

Round 1
Reviewer 1 Report
Comments and Suggestions for Authors
This study investigates the combined effects of different water conditions and biochar on tomato physiology, growth, yield, and water use efficiency. Although similar studies have been reported to some extent, the authors used tomato, a water-sensitive crop, as the plant material and conducted the experiment in Saudi Arabia, which can be considered an innovation in the application context and holds significant practical value. The overall writing of the article is relatively fluent; the research methods are classical and reliable; and the results support the conclusions. The authors concluded that under water-deficient conditions, the application of biochar can improve water use efficiency and alleviate water stress. In my opinion, this article can be reconsidered after major revisions. The main issues with the article are as follows:
(1) The authors placed the Materials and Methods section in the third part, after the Results, which is not a common practice in our papers. Please revise it according to the requirements of the Plants journal. Additionally, there are several instances of lack of detail in the text, such as in section 3.1, where "experience" is used instead of "experiment." The author should make corresponding revisions and check the entire text.
(2) The authors omitted an important section on sampling. It is unclear from the article how many samples were collected, nor is it possible to determine how many samples were averaged and calculated to derive the values in the results.
(3) A significant flaw in the article is the absence of standard deviations, with only significance levels provided. The authors are advised to reorganize the data and include standard deviations in the charts.
(4) The Discussion section includes content related to research significance and background, which should not be included in this part. The discussion should be concise, with direct and clear conclusions, and should not be overly verbose.
(5) In the Results section, the analysis methods employed by the authors are overly simplistic, focusing only on comparisons between treatments.
In summary, this paper has relatively good significance and draws clear research conclusions, offering considerable value for production and making a certain academic contribution. However, the most critical issue is the lack of standard deviations, which reduces the credibility of the data. The authors are recommended to revise the manuscript and resubmit it.
Author Response
Reviewer # 1
This study investigates the combined effects of different water conditions and biochar on tomato physiology, growth, yield, and water use efficiency. Although similar studies have been reported to some extent, the authors used tomato, a water-sensitive crop, as the plant material and conducted the experiment in Saudi Arabia, which can be considered an innovation in the application context and holds significant practical value. The overall writing of the article is relatively fluent; the research methods are classical and reliable; and the results support the conclusions. The authors concluded that under water-deficient conditions, the application of biochar can improve water use efficiency and alleviate water stress. In my opinion, this article can be reconsidered after major revisions. The main issues with the article are as follows:
Response: Thank you for your encouragement.
Comment (1) The authors placed the Materials and Methods section in the third part, after the Results, which is not a common practice in our papers. Please revise it according to the requirements of the Plants journal. Additionally, there are several instances of lack of detail in the text, such as in section 3.1, where "experience" is used instead of "experiment." The author should make corresponding revisions and check the entire text.
Response: We appreciate the reviewer’s comment. According to the official Plants journal template available on the MDPI website, the Materials and Methods section is indeed positioned after the Results section. Therefore, we have aligned the manuscript with the journal’s format. Additionally, the rest of the comment has been addressed, and the relevant revisions have been incorporated throughout the manuscript.
Comment (2) The authors omitted an important section on sampling. It is unclear from the article how many samples were collected, nor is it possible to determine how many samples were averaged and calculated to derive the values in the results.
Response: Thank you very much for your suggestion. The number of samples and treatments was added to the text.
Comment (3) A significant flaw in the article is the absence of standard deviations, with only significance levels provided. The authors are advised to reorganize the data and include standard deviations in the charts.
Response: We appreciate the reviewer’s valuable comment. The standard error bars have been added to all relevant figures to provide a clearer representation of data variability.
Comment (4) The Discussion section includes content related to research significance and background, which should not be included in this part. The discussion should be concise, with direct and clear conclusions, and should not be overly verbose.
Response: The first two sentences in the results and discussion were omitted as they might be related to the background of the effect of water stress. Therefore, it was removed.
Comment (5) In the Results section, the analysis methods employed by the authors are overly simplistic, focusing only on comparisons between treatments.
Response: The current work takes into account the single effect of the biochar and water stress for the yield and water productivity (Table 1). In addition, the effect of both treatments on the interaction and the comparison between the treatment of Biochar and water stress.
In summary, this paper has relatively good significance and draws clear research conclusions, offering considerable value for production and making a certain academic contribution. However, the most critical issue is the lack of standard deviations, which reduces the credibility of the data. The authors are recommended to revise the manuscript and resubmit it.
Response: Thank you again.
Reviewer 2 Report
Comments and Suggestions for Authors
In this manuscript, Abdullah Obadi and colleagues examined the impacts of biochar addition on morpho-physiological characteristics, yield, and water productivity (WP) in greenhouses under drought-stress conditions. I have following comments:
1, The present title is suitable for a Review but not research paper. I suggest to employ “Analysis of Biochar Addition in Improving Tomato Morpho-Physiological Traits and Water Productivity in Greenhouse”.
2, For the Abstract, too many values are employed. For instance, the sentences “For instance, irrigation water shortage decreased yield by 5.91%, 16.16%, and 30.88% at 80%, 60%, and 40% of ETc, respectively, compared to (100% of ETc). However, DI improved WP by 17.61%, 39.76%, and 72.80% at 80%, 60%, and 40% of ETc, respectively, compared to the control. The interaction between DI and BC positively affected morphological, physiological, yield, and WP. For instance, the highest rate of biochar (BC 5%) increased yield by 4.59%, 11.92%, 4.96%, and 3.50% at 100%, 80%, 60%, and 40% of ETc, respectively, compared to untreated plants (BC0%). ” contain too many detailed data, which should be summarized in the Abstract.
3, For the introduction, more citation should be provided. For instance, Line 85-87 contains no citation.
4, For the results, error bar should be exhibited in Figures 1, 2, 3, and 4, and error value should be provided in Tables 1, 2, and 3.
5, For the materials and methods, plant growth conditions like temperature and humidity should be provided, and randomization in the samplings should be clearly described.
6, An independent discussion section should be included.
Author Response
Comment (1) The present title is suitable for a Review but not research paper. I suggest to employ “Analysis of Biochar Addition in Improving Tomato Morpho-Physiological Traits and Water Productivity in Greenhouse”.
Response: The title has been changed to “Analysis of Biochar Addition in Improving Tomato Morpho-Physiological Traits and Water Productivity in Greenhouse” as suggested by the reviewer, which is shorter and more reflective of the work.
Comment (2) For the Abstract, too many values are employed. For instance, the sentences “For instance, irrigation water shortage decreased yield by 5.91%, 16.16%, and 30.88% at 80%, 60%, and 40% of ETc, respectively, compared to (100% of ETc). However, DI improved WP by 17.61%, 39.76%, and 72.80% at 80%, 60%, and 40% of ETc, respectively, compared to the control. The interaction between DI and BC positively affected morphological, physiological, yield, and WP. For instance, the highest rate of biochar (BC 5%) increased yield by 4.59%, 11.92%, 4.96%, and 3.50% at 100%, 80%, 60%, and 40% of ETc, respectively, compared to untreated plants (BC0%). ” contain too many detailed data, which should be summarized in the Abstract.
Response: The abstract has been shortened, and unnecessary data have been omitted.
Comment (3) For the introduction, more citation should be provided. For instance, Line 85-87 contains no citation.
Response: Thank you for this valuable comment. Many cited references have been added.
Comment (4) For the results, error bar should be exhibited in Figures 1, 2, 3, and 4, and error value should be provided in Tables 1, 2, and 3.
Response: We appreciate the reviewer’s valuable comment. The standard error bars have been added to all relevant figures and tables to provide a clearer representation of data variability.
Comment (5) For the materials and methods, plant growth conditions like temperature and humidity should be provided, and randomization in the samplings should be clearly described.
Response: Thank you for this valuable comment.
Regarding the first part of your suggestion, information about plant growth conditions (including temperature and humidity) has now been added to the Materials and Methods section as requested.
Concerning the second part of your comment, the details of randomization and the experimental design were already included in the manuscript under the Materials and Methods section, as follows:
“The experiment was conducted in a split-plot and randomized complete block design with three replications. The main plots irrigation water levels were allocated with three levels of deficit irrigation water (40, 60, and 80%, based on evapotranspiration (ETc)) and full irrigation (100% of ETc), and soil amendments as biochar by two levels (3 and 5% w/w) denoted as (BC3% and BC5%) in addition to the control, respectively, randomly distributed as sub-pieces. The experiment included 36 units [4 irrigation levels × 3 biochar levels × 3 replicates]. The experimental unit was a 6 m long and 1 m wide line with an irrigation line spacing of 0.40 m between emitters. It contained 15 plants. The 100% of ETc treatment without biochar additives was the control (Fig. 7).”
We believe this description provides an adequate clarification of the randomization procedure and experimental layout.
Comment (6) An independent discussion section should be included.
Response: Thank you for your valuable suggestion. We have intentionally adopted the combined “Results and Discussion” format in this manuscript. This approach aligns with the publication style of many articles in Plants, where results and their interpretation are presented together to maintain a logical flow and avoid redundancy. We have also followed this format in our previous publications in the same journal. Therefore, we have retained the current structure but further enriched the discussion to enhance the depth of interpretation and clarity of findings.
Round 2
Reviewer 1 Report
Comments and Suggestions for Authors
The MS has been much improved. I think it can be published in present form.
Reviewer 2 Report
Comments and Suggestions for Authors
Authros have addressed my concerns in the revision.